

# Fungal infection, decline and persistence in the only obligate troglodytic Neotropical salamander

Mizraim Olivares-Miranda[1], Vance T. Vredenburg[2], Julio C. García-Sánchez[1], Allison Q. Byrne[3], Erica B. Rosenblum[3] and Sean M. Rovito[1]

[1] Unidad de Genómica Avanzada (Langebio), Centro de Investigación y de Estudios Avanzados del Instituto Politécnico Nacional, Irapuato, Guanajuato, México
[2] Department of Biology, San Francisco State University, San Francisco, CA, USA
[3] Department of Environmental Science, Policy, and Management, University of California, Berkeley, Berkeley, CA, USA

Corresponding author
Sean M. Rovito,
sean.rovito@cinvestav.mx

## ABSTRACT

The fungal pathogen *Batrachochytrium dendrobatidis* (*Bd*) is implicated in global mass die-offs and declines in amphibians. In Mesoamerica, the *Bd* epidemic wave hypothesis is supported by detection of *Bd* in historic museum specimens collected over the last century, yet the timing and impact of the early stages of the wave remain poorly understood. *Chiropterotriton magnipes*, the only obligate troglodytic Neotropical salamander, was abundant in its small range in the decade following its description in 1965, but subsequently disappeared from known localities and was not seen for 34 years. Its decline is roughly coincident with that of other populations of Neotropical salamanders associated with the invasion and spread of *Bd*. To determine the presence and infection intensity of *Bd* on *C. magnipes* and sympatric amphibian species (which are also *Bd* hosts), we used a noninvasive sampling technique and qPCR assay to detect *Bd* on museum specimens of *C. magnipes* collected from 1952 to 2012, and from extant populations of *C. magnipes* and sympatric species of amphibians. We also tested for the presence of the recently discovered *Batrachochytrium salamandivorans* (*Bsal*), another fungal chytridiomycete pathogen of salamanders, using a similar technique specific for *Bsal*. We did not detect *Bd* in populations of *C. magnipes* before 1969, while *Bd* was detected at low to moderate prevalence just prior to and during declines. This pattern is consistent with *Bd*-caused epizootics followed by host declines and extirpations described in other hosts. We did not detect *Bsal* in any extant population of *C. magnipes*. We obtained one of the earliest positive records of the fungus to date in Latin America, providing additional historical evidence consistent with the *Bd* epidemic wave hypothesis. Genotyping results show that at least one population is currently infected with the Global Panzootic Lineage of *Bd*, but our genotyping of the historical positive samples was unsuccessful. The lack of large samples from some years and the difficulty in genotyping historical Bd samples illustrate some of the difficulties inherent in assigning causality to historical amphibian declines. These data also provide an important historical baseline for actions to preserve the few known remaining populations of *C. magnipes*.

salamandivorans, Chytridiomycosis, Caves, Mexico

## INTRODUCTION

Chytridiomycosis, a fungal disease caused by *Batrachochytrium dendrobatidis* (*Bd*) and
*Batrachochytrium salamandivorans (Bsal)*, is recognized as a driver of amphibian
population declines and extirpations on a global scale, which many believe is indicative of a
sixth mass extinction on earth (*Wake & Vredenburg, 2008*; *Scheele et al., 2019*).
One particular lineage of *Bd*, the Global Panzootic Lineage (Bd-GPL), is particularly
virulent and associated with most documented declines (*Farrer et al., 2011*). In many of the
most affected areas, such as Central America, South America, and Australia, Bd-GPL
appears to have been historically absent (prior to the 1970s) and amphibian declines
followed after *Bd* invaded and rapidly increased in prevalence and in geographical
distribution (*Cheng et al., 2011*; *Lips, 2016*). Despite the apparent recent and rapid spread
of Bd-GPL to many parts of the world, recent genomic studies indicate that some areas
(including South America) may have long harbored other endemic lineages of *Bd*
(*O'Hanlon et al., 2018*; *Byrne et al., 2019*), complicating interpretation of declines when
direct evidence of chytridiomycosis-associated mortality is lacking.

Latin America, home of tremendous amphibian diversity, has some of the most
*Bd*-impacted populations in the world, with major declines in both frogs and salamanders
at multiple sites from the 1970s onward (*Young et al., 2001*; *La Marca et al., 2005*;
*Rovito et al., 2009*; *Catenazzi et al., 2011*). The earliest published records of *Bd* in
Mesoamerica are from Costa Rica in the 1960s, predating *Bd* epizootics in that country,
and from Mexico in the early 1970s where its arrival was temporally coincident with major
declines in salamanders (*Rovito et al., 2009*; *Cheng et al., 2011*; *De León et al., 2019*).
About two decades later, *Bd*-epizootics were directly documented in southeastern Costa
Rica and Panama (*Berger et al., 1998*; *Lips, 1998*; *Lips et al., 2006*) and provided some of the
foundational data used to describe *Bd* in 1999 (*Longcore, Pessier & Nichols, 1999*).

The epidemic wave hypothesis (*Lips et al., 2006*, *2008*) states that Bd-GPL moved
through naïve amphibian host species, causing massive population declines and
extirpations. Currently available data on declines from Latin American in the late 20th
century, bolstered by retrospective museum surveys, generally support this hypothesis, but
many key elements of this paradigm remain incompletely understood. According to one
hypothesized sequence of events, *Bd* invaded Mexico and Guatemala first and spread
southeast, yet despite many species that have declined in Mexico and Guatemala, relatively
few historical amphibian declines have been linked to *Bd* (*Cheng et al., 2011*; *Mendelson
et al., 2014*). Furthermore, recent work on *Bd* genomics (*O'Hanlon et al., 2018*; *Byrne et al.,
2019*) reveals that multiple *Bd* lineages can coexist in some regions, and may have
differential impact on host health. A *Bd*-positive frog specimen from the Bolivian Andes
collected in 1863, long before declines of cloud forest amphibians in that country in the
early 1990s, shows that *Bd* has been present in South America long before reported

epizootics. This early positive record was detected by qPCR, and histological examination was suggestive but inconclusive (*Burrowes & De la Riva, 2017*). In cases of very early detection of *Bd* in countries where declines occurred much later, it is possible that early detections correspond to endemic strains while declines were caused only by the arrival of the more virulent Bd-GPL, but genetic data to support this hypothesis are currently lacking (*Burrowes & De la Riva, 2017*; *Burrowes et al., 2020*). Retrospective surveys of key species in museum collections, together with the use of genomic methods to identify *Bd* lineage, offer a promising avenue to fill gaps in our knowledge related to the arrival and impact of *Bd* in the Neotropics.

Compared to other Neotropical countries, Mexico is particularly rich in salamander species, with the second highest number of species of any country (158 described species; *AmphibiaWeb, 2020*). The large majority of these species belong to a single family, the Plethodontidae, which has experienced an evolutionary radiation in the mountains of Mexico. Among its many endemic plethodontids, Mexico has the only troglodytic (cave-dwelling) salamander species in the Neotropics. The single obligate troglodytic species, *Chiropterotriton magnipes*, was described in 1965 (*Rabb, 1965*) and was known from a small number of caves in the states of Querétaro and San Luis Potosí. Its unique morphology (massive fully webbed feet, protruding eyes, mostly unpigmented skin) make it unique among Neotropical salamanders, and thus of high evolutionary significance. It was initially abundant at several localities and is well represented in museum collections from the 1960s to 1970s, but later became scarce and was not seen from 1982 to 2005 (*Parra-Olea, Wake & Hanken, 2008*). In recent years, it has been found at a small number of localities in low abundance compared to the decade following its description (S. Rovito, 2017, personal observation). Because most cave habitat at known sites apparently remains largely intact (although not without human impact), the causes of decline are not apparent, and thus infectious disease could be involved. Understanding the cause of past declines is thus critical to designing effective conservation strategies to ensure its persistence.

While the approximate timing of declines coincides with those seen at other localities in central Mexico (*Cheng et al., 2011*), the range of *C. magnipes* is farther north than any of these localities. The large number of specimens from the pre-decline period present in museums, together with a small number of specimens from the period of decline, allowed us to test whether the timing of arrival of *Bd* (if present) in these populations coincides with their declines. Thus, we combine retrospective surveys of nearly all specimens of this species in museum collections to test this hypothesis. We also use recently developed genotypic methods that employ small amounts of DNA to genotype present day *Bd*-positive samples from remaining populations of *C. magnipes* and we attempt to genotype historical *Bd*-positive specimens to identify which *Bd* lineage might have been involved in declines. By understanding the role of *Bd* in historical declines and its current impact on remnant populations, we suggest a conservation strategy for this threatened and emblematic Mexican salamander.

## MATERIALS AND METHODS

### Historical prevalence of *Bd* in museum specimens of *C. magnipes*

We obtained skin swab samples from 279 formalin-preserved museum specimens of *C. magnipes* collected in Mexico from 1952 to 2012. Our sample included six specimens identified only as *Chiropterotriton* sp. from a historical locality of *C. magnipes* collected in 1981 and 1982. The specimens are housed at the University of Michigan Museum of Zoology (UMMZ), Museum of Vertebrate Zoology, University of California, Berkeley (MVZ), Biodiversity Research and Teaching Collections, Texas A&M University (TCWC), Field Museum of Natural History (FMNH), Museum of Comparative Zoology, Harvard University (MCZ) and the Louisiana State University Museum of Zoology (LSUMZ). Before sampling, each specimen was rinsed with 70% EtOH to avoid contamination by errant skin pieces or floating *Bd* zoospores possibly contained in the preservation jars. Multiple specimens (sometimes over 100) of *C. magnipes* were stored in a single jar, particularly in the MVZ where the bulk of our sample is from. While rinsing specimens should reduce the probability of cross contamination, transfer of *Bd* DNA or zoospores between samples within a jar cannot be excluded using our methodology. Using nitrile gloves and sterile Medical Wire Dryswabs (MW113; Medical Wire & Equipment Co, Corsham, UK), we swabbed each specimen 30 times (10 strokes for the ventral surface and five strokes per ventral part of the feet). Nitrile gloves were changed between specimens and the swabs were air dried and stored at −20 °C until processing.

We extracted DNA following the standard extraction protocol for PrepMan Ultra (Applied Biosystems, Foster City, CA, USA). All the extractions were diluted in 0.25X TE buffer and used as template for the quantitative Polymerase Chain Reaction (qPCR) assay with modifications for detection of zoospore genomic equivalents (GE) of *Bd* in historical samples (*Cheng et al., 2011*). We used the primers and probe developed by *Boyle et al. (2004)* and the ABI 7300 real-time PCR system at San Francisco State University to amplify the genomic region ITS3–5.8S of *Bd*. We ran the samples along with *Bd*-positive controls at the dilution levels of 0.1, 1.0, 10 and 100 GE (three positive controls per dilution level per plate) and negative controls (water and TE buffer, four total per plate) to detect false positives. All samples were run in triplicate and were considered as *Bd*-positive when one of the three reactions returned a positive quantification (>0 GE) and when the reaction curve of the positive samples showed an exponential tendency according to the *Bd* positive controls, with the standard reaction curves generated having a $R^2$ value > 0.99. Because this qPCR assay may not return accurate values of *Bd*-infection intensity for historical formalin-preserved *Bd*-positive samples (*Cheng et al., 2011*), the results of the qPCR assay were treated only as *Bd*-positive or negative.

### Current prevalence of *Bd/Bsal*

We conducted searches for *C. magnipes* at four historical sites, two recently discovered populations, and eight caves where salamanders had not previously been reported (Table 1; Fig. 1). Searches consisted of scanning cave walls, ceiling and floor with headlamps until the entire accessible area had been surveyed. Each cave was searched

**Table 1 Locality information and number of individuals detected for all live *Chiropterotriton magnipes* tested for *Bd*.** Locality information, geographic coordinates, and number of individuals of *C. magnipes* detected for caves where searches for *C. magnipes* were conducted. Locality numbers correspond to those in Fig. 1. Locality names with an asterisk are historical localities that are represented in museum collections. Number of *C. magnipes* detected is greater than number of individuals sampled for *Bd* in some cases because not all salamanders detected were accessible for sampling. For *C. magnipes* detected, letters in parentheses indicate adult females (F), adult males (M), juveniles (J), or individuals not captured (N). Subscript C refers to voucher specimens collected. Geographic coordinates provided to only two decimal places to protect sensitive cave habitat.

| Locality number | Locality name | State | Latitude | Longitude | Search date | Number and age class/sex of *C. magnipes* detected |
|---|---|---|---|---|---|---|
| 1 | Cueva de los Cuartos | Qro | 21.34 | −99.15 | 8 July 2017 | 0 |
| 2 | Cueva de los Lirios | Qro | 21.39 | −99.14 | 8 July 2017 | 0 |
| 3 | Lagunitas de San Diego | Qro | 21.37 | −99.14 | 25 Mar 2017 | 0 |
| 4 | 3.4 km W of El Madroño, Qro* | Qro | 21.26 | −99.18 | 14 Sept 2017 | 0 |
| 5 | Cueva de la Araña, El Madroño* | Qro | 21.27 | −99.16 | 24 Mar 2017, 6 July 2017, 7 July 2017, 25 Aug 2017, 10 Oct 2018 | 0 |
| 6a | Llano de los Caballos | SLP | 21.40 | −99.08 | 6 July 2017 | 0 |
| 6b | Cueva de los Cabritos | SLP | 21.40 | −99.08 | 6 July 2017 | 0 |
| 7 | La Trinidad | SLP | 21.40 | −99.07 | 6 July 2017, 7 July 2017, 1 Aug 2018, 24 Aug 2017, 18 Oct 2018 | 3 ($F_C$, N, N), 4 (F, F, J, J), 1 (M), 1 (F), 2 (M, F) |
| 8 | Cueva Llano Chico | SLP | 21.38 | −99.10 | 2 Aug 2017 | 0 |
| 9 | Cueva de las Golondrinas, San Antonio | SLP | 21.36 | −99.01 | 26 Mar 2017 | 0 |
| 10 | Cueva de la Barranca | SLP | 21.33 | −99.05 | 26 Mar 2017, 7 July 2017 | 0 |
| 11 | Cueva de la Iglesia, Ahuacatlán* | SLP | 21.32 | −99.05 | 23 Mar 2017 | 0 |
| 12 | Cueva de Potrerillos (type locality) | SLP | 21.31 | −99.07 | 23 Mar 2017, 6 July 2017 | 0 |
| 13 | Cueva El Coni, Durango | Hgo | 20.89 | −99.23 | 21 Mar 2017, 12 June 2017, 31 July 2017, 23 Aug 2017, 8 Oct 2018, 17 Oct 2018 | 1 (M), 1 ($F_C$), 3 (F, F, F), 1 (M), 0, 0 |

between one and six times, depending on its assessed potential for salamander presence (Table 1). Surveys and manipulation of amphibians were conducted in accordance with the guideline of the Institutional Care and Use of Laboratory Animals Committee of the SIACUAL System of Cinvestav, Mexico (Protocol 0291-19 to Sean Rovito). Permits for field research were provided by the Secretaría del Medio Ambiente y Recursos Naturales (SEMARNAT permit SGPA/DGVS/002764/18 to Sean Rovito).

In order to estimate the prevalence of *Bd* in the few caves where we found extant populations of *C. magnipes*, we obtained skin swabs from live *C. magnipes* from two

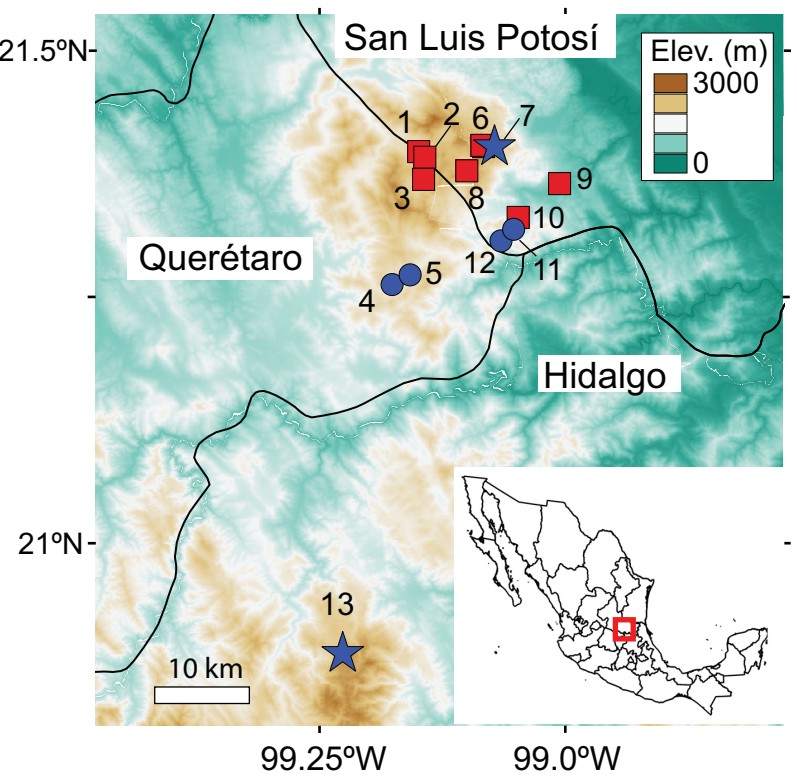

**Figure 1 Map of caves where searches for *C. magnipes* were conducted.** Historical sites where *C. magnipes* was collected are shown as blue circles, sites where *C. magnipes* was recorded during our surveys as blue stars, and caves where *C. magnipes* was not previously reported and was not detected in our survey as red squares. Locality numbers correspond to those in Table 1. Historic and new records of *C. magnipes* encompass entire range of the species, and no populations are known between population 13 in Hidalgo and the rest of the species' range.

localities of Mexico (La Trinidad, San Luis Potosí and El Coni, Durango, Hidalgo; Table 1; Fig. 1) in 2017 and 2018. We did not mark animals to avoid compromising their health in any way but did note age class (adult/juvenile) and sex to determine the minimum number of individuals sampled. We also took skin swabs from two salamander and two frog species found in sympatry in caves with *C. magnipes*: *Chiropterotriton mosaueri*, *Aquiloeurycea* sp., *Craugastor decoratus* and *Eleutherodactylus cystignathoides*. With the exception two juvenile *C. magnipes* sampled at La Trinidad, all salamanders and frogs tested for *Bd* were adults or subadults. We captured each individual in a clean plastic bag inside the cave. Each individual amphibian was rinsed with 100 mL of sterile water and nitrile gloves were changed between specimens to avoid cross contamination. We use sterile a Medical Wire Dryswab (MW113) to swab each individual 30 times using a standard sampling protocol (*Hyatt et al., 2007*). Individuals were released immediately after being swabbed. We obtained 29 skin swab samples, which were stored in 1.5 mL cryogenic tubes in liquid nitrogen for transport to the lab, and subsequently at −80 °C until processing.

We extracted DNA from the 29 skin swab samples, diluted and ran the qPCR assay as for *Bd* as described in the previous section. A second qPCR assay for *Bsal* was performed in
the Catenazzi Lab at the Florida International University using the primers and probe developed by *Blooi et al. (2013)*, adding to the reaction plate *Bsal* positive controls and negative controls to detect false positives. All samples were run in duplicate and were regarded as *Bd* or *Bsal* positive when the two duplicates returned a positive quantification (>0 GE) and when the reaction curve of the positive samples shows an exponential tendency according to the *Bd/Bsal* positive controls, with the standard reaction curves generated having a $R^2$ value > 0.99. The mean quantity output data of the positive qPCR results were multiplied by 80 (to account the dilution factor) to obtain the mean infection intensity of the swab collected for each animal.

## Genotype of positive samples for *Bd*

We used a multiplexed microfluidic PCR approach (*Byrne et al., 2017*) to attempt to genotype Bd from six historical museum swabs and two field-collected swabs (from El Coni, Hidalgo) of *C. magnipes* that tested positive for *Bd* using the qPCR assay. Briefly, this approach uses an isopropanol precipitation followed by a pre-amplification enrichment step. Samples are then amplified at 192 loci selected to discriminate between Bd lineages. Loci were amplified in a Fluidigm Access Array (Fluidigm, South San Francisco, CA, USA) and sequenced on an Illumina MiSeq v3 sequencer at The University of Idaho IBEST Genomics Resources Core. This method has been shown to work well when DNA extracts from swab samples and accurately discriminates between known *Bd* lineages (*Byrne et al., 2019*). The multilocus sequence data used for Bd genotyping are accessible via the NCBI Short Read Archive (BioProject ID PRJNA608001).

Raw sequence reads were processed as described in *Byrne et al. (2017)*. Briefly, reads were joined using FLASH (v.1.2.11) (*Magoc & Salzberg, 2011*) and consensus sequences were produced from sequence variants represented by at least 5 reads and 5% of total reads for that sample/amplicon combination. Multiple alleles at an individual locus were represented using IUPAC ambiguity codes. To determine what *Bd* lineage was represented in the available samples, we used a gene tree to species tree approach. First, we selected newly genotyped samples with at least 20 amplicon sequences (N = 1). Next, we combined our newly genotyped sample with 105 previously published globally distributed samples (*Byrne et al., 2017*; *O'Hanlon et al., 2018*). We selected amplicons with no more than 50% missing data (N = 185), individually aligned all amplicon sequences using MUSCLE package (v.3.18.0) (*Edgar, 2004*) in R (v.3.4.3) (*R Development Core Team, 2015*) and produced gene trees using RAxML (v.8.2.11) (*Stamatakis, 2014*) to search for the best-scoring ML tree from 100 bootstrap replicates. We collapsed all nodes with less than 10 bootstrap support and used these collapsed gene trees as input to Astral-III (v.5.5.9) (*Zhang et al., 2018*). Astral estimates an unrooted species tree given unrooted gene trees and allows for missing data across gene trees. Finally, we collapsed all nodes in the Astral species tree with a posterior less than 0.7.

## Statistical analysis

For the historical specimens, *Bd* prevalence for each year was calculated dividing the total of *Bd* positive *C. magnipes* specimens by the total of specimens sampled. For years with

**Table 2 Results of qPCR assay for *Bd* in historical museum samples by year sampled.**

| Year of collection | Localities sampled (no. sampled) | No. sampled | No. positives for *Bd* | % *Bd* prevalence (95% clopper–pearson confidence interval) |
|---|---|---|---|---|
| 1952 | W of El Madroño, Qro (1), Cueva de Potrerillos, SLP (2) | 3 | 0 | 0 [0–60] |
| 1955 | Cueva de Potrerillos, SLP (1) | 1 | 0 | 0 [0–84] |
| 1964 | Cueva de Potrerillos, SLP (46) | 46 | 0 | 0 [0–8] |
| 1965 | El Sotano, SLP (4) | 4 | 0 | 0 [0–52] |
| 1968 | W of El Madroño, Qro (40) | 40 | 0 | 0 [0–13] |
| 1969 | W of El Madroño, Qro (38), Cueva de Potrerillos, SLP (43) | 81 | **1** | **1.2 [0–7]** |
| 1970 | W of El Madroño, Qro (46) | 46 | **1** | **2.2 [0–12]** |
| 1971 | W of El Madroño, Qro (10) | 10 | 0 | 0 [0–26] |
| 1975 | Cueva de la Iglesia, SLP (33) | 33 | **3** | **9.1 [3–24]** |
| 1979 | W of El Madroño, Qro (3) | 3 | 0 | 0 [0–60] |
| 1981 | W of El Madroño, Qro (1), Cueva de la Iglesia, SLP (1) | 2 | **1** | **50 [9–91]** |
| 1982 | Cueva de la Iglesia, SLP (5), Cueva de Potrerillos, SLP (4) | 9 | **1** | **11.1 [3–44]** |
| 2012 | Cueva el Coni, Hgo (1) | 1 | 0 | 0 [0–84] |
| **Total** | | **279** | **7** | **2.5 [1–5]** |

Note:
Bold font indicates samples that tested positive for *Bd*. Numbers in parentheses after locality name indicate number of museum specimens swabbed from the locality from each year. Various localities within a few km west of El Lobo and El Madroño, Qro are listed as "W of El Madroño".

zero *Bd* prevalence, we calculate the probability of a false negative using the formula $(1 - 0.05)^S$, where $S$ represents the sample size of the year and taking in count a true *Bd* prevalence of 0.05 in the population (*Cheng et al., 2011*). Current *Bd* and *Bsal* prevalence for each species and locality was calculated dividing the total of *Bd/Bsal* positive individuals by the total of individuals sampled. We then calculated 95% confidence intervals using the Clopper-Pearson test (also known as an exact binomial distribution confidence interval test) using the package "PropsC". All statistical analyses were performed in R version 3.4.1.

# RESULTS

## Historical *Bd* prevalence in populations of *C. magnipes*

We found a pattern of emergence of *Bd* in the study consistent with the arrival of *Bd* to naïve amphibian populations. From 1952 to 1968, all samples were negative for *Bd* ($N = 91$), and all positive samples were found in the period 1969–1982. Considering all the historical preserved specimens of *C. magnipes* sampled, we found a *Bd* prevalence of 2.5% (7/279) (Table 2). The earliest record of a *Bd*-positive animal came from a single specimen collected in 1969, representing 1.2% (1/81) of the total sampled specimens from that year. We obtained *Bd*-positive specimens in subsequent years (1970, 1975, 1981 and 1982). For the period prior to the first detection of *Bd* (1952–1968) and for the years with relatively large sample sizes (1964 and 1968), we calculated the probability of a false negative (assuming that *Bd* had been present at a prevalence of 5%). The probability of a false negative was 0.007 for 1952–1968, 0.09 for 1964 and 0.12 for 1968.

**Table 3 Species and locality information, sample sizes, and results of qPCR *Bd* and *Bsal* assays for amphibians encountered during fieldwork.**

| Species | Locality | N Bd | Bd+ | % Bd prevalence (95% CI) | Mean infection intensity (GE) | N Bsal | Bsal+ | % Bsal prevalence (95% CI) |
|---|---|---|---|---|---|---|---|---|
| Salamanders | | | | | | | | |
| *Chiropterotriton magnipes* | El Coni, Hgo | 6 | **2** | **33 [4–78]** | **123.5** | 4 | 0 | 0 [0–60] |
| *Chiropterotriton magnipes* | La Trinidad, SLP | 9 | **2** | **22 [3–60]** | **392** | 5 | 0 | 0 [0–52] |
| *Chiropterotriton mosaueri* | El Coni, Hgo | 3 | **1** | **33 [1–90]** | **2517.7** | 2 | 0 | 0 [0–84] |
| *Aquiloeurycea* sp. | El Coni, Hgo | 6 | **2** | **33 [4–78]** | **622.8** | 3 | 0 | 0 [0–70] |
| Frogs | | | | | | | | |
| *Craugastor decoratus* | Cueva la Barranca, SLP | 2 | **1** | **50 [13–98]** | **1779.5** | — | — | — |
| *Eleutherodactylus cystignathoides* | El Coni, Hgo | 2 | 0 | 0 [0–84] | — | — | — | — |
| *Eleutherodactylus cystignathoides* | Cueva de la Iglesia, SLP | 1 | 0 | 0 [0–97] | — | — | — | — |
| **Total** | | **29** | **8** | **27.6 [13–47]** | | **14** | **0** | **0 [0–23]** |

**Note:**
Bold font indicates samples that tested positive for *Bd*.

## Current *Bd* and *Bsal* prevalence in caves of Mexico

A total of 29 skin swabs of salamanders and frogs were collected from cave study sites; we did not observe dead amphibians or individuals with skin lesions at any site. *Chiropterotriton magnipes* was found in two of the 15 sites surveyed, with both extant populations representing new localities; no individuals were found in sites with historical records of the species. Of the six *C. magnipes* we found at El Coni, Hidalgo, we sampled a minimum of five different individuals based on age and sex of the specimens and of the nine *C. magnipes* from La Trinidad, we sampled a minimum of six different individuals (Table 1). We detected *Bd* at three of the six sites where we found amphibians (Table 3) with 27.6% (8/29) of individuals tested being *Bd*-positive. Both of the sites we surveyed with extant populations of *C. magnipes* (El Coni, Hidalgo and La Trinidad, San Luis Potosí) tested positive for *Bd*, with a prevalence of 29.1% (7/24) and a low to moderate infection intensity (6–2,618 GE) across species at El Coni, and a lower prevalence (11%, 2/9 samples) and relatively low infection intensity (312–472 GE) at La Trinidad. ITS copy number can vary between *Bd* strains, affecting prevalence and infection intensity estimates from qPCR (*Longo et al., 2013*; *Rebollar et al., 2017*). Because we did not have a *Bd* isolate from these populations to quantify ITS copy number, estimates of GE should be regarded as approximate. *Bd*-positive individuals of *C. magnipes* were detected at both sites (Table 3). Other salamander species found in caves with extant populations of *C. magnipes* tested positive for *Bd*, with 1/3 *Chiropterotriton mosaueri* and 2/6 *Aquiloeurycea sp.* testing positive. A single frog (*Craugastor decoratus*) from a cave where no *C. magnipes* were found also tested positive for *Bd*. All individuals that tested positive for *Bd* were adults or subadults. Of the 14 salamanders included in our sample, none tested positive for *Bsal*.

### *Bd* genotyping

Genotyping was successful for only one of the two modern *Bd*-positive samples of *C. magnipes* from El Coni, Hidalgo and matched the Global Panzootic Linage subclade 1 (Bd-GPL-1). In our phylogenetic analyses we found that the sample successfully sequenced was most closely related to another sample collected from Chiapas, Mexico and published previously (*Byrne et al., 2019*) (Fig. S1). None of the six historical *Bd*-positive samples produced any usable sequence data for genotyping.

## DISCUSSION

Detection of *Bd* in extant populations of *C. magnipes*, together with its apparent arrival shortly before observed population declines based on museum samples, point to a role of *Bd* in the declines of this species. Other aspects of our results, including relatively low *Bd* prevalence during declines, make the overall picture involving the role of *Bd* in the decline of *C. magnipes* less clear. While we detected Bd-GPL in the one extant sample we successfully genotyped, our failure to successfully genotype historical positives complicates interpretation of our historical *Bd* results.

Our retrospective historical survey of *Bd* prevalence in populations of *C. magnipes* revealed that *Bd* was not detected in a large sample size from 1952 to 1968, during a period of high salamander abundance, and was detected shortly before and during the subsequent period of decline in salamander abundance. The estimate of *Bd* prevalence for the final year with a relatively large sample of specimens (1975, 9.1% (95% CI [3–24])) is higher than in previous years following first detection (0–2.2% prevalence, upper limit of 95% CI 11%). These results are consistent with the arrival of a novel pathogen to populations of a naïve host, as outlined in the epidemic wave hypothesis (*Lips et al., 2008*). Other aspects of our results are less consistent with a *Bd*-associated decline, however. Following first detection, the prevalence of *Bd* remained low to moderate in our sample (1.2–50% between 1969 and 1982), and *Bd* was not detected our relatively small samples from 1971 (10 specimens) and 1979 (three specimens). Furthermore, the 95% confidence intervals for *Bd* prevalence overlap for all years the fungus was detected. Despite our near-complete sampling of available museum specimens, the small sample size for all years following 1975 (resulting directly from declines in salamander abundance) make it difficult to infer trends in *Bd* prevalence with confidence.

Our contemporary surveys for *Bd* show that the fungus is present in both sites with extant *C. magnipes* populations, although not at especially high prevalence (Table 3). Individuals of *C. magnipes*, (2/6 at El Coni and 2/9 at La Trinidad) tested positive for *Bd* and two of the other three species that we encountered at El Coni had *Bd*-positive individuals. Only *Eleutherodactylus cystignathoides* from El Coni tested negative for *Bd* for all samples (although we collected only two samples of this species), and only one individual of any other species (a *Craugastor decoratus*) tested positive from the four other caves where we detected amphibians. Overall, the prevalence of *Bd* in our contemporary sample was relatively low (27.6%), roughly comparable to that of Australian amphibians in enzootic stages of *Bd* infection surveyed >8 years after declines (15–28%) (*Retallick, McCallum & Speare, 2004*; *Woodhams & Alford, 2005*; *Kriger & Hero, 2006*). The one

contemporary sample that we were able to genotype was assigned to the Global Panzootic Lineage, which has been implicated in amphibian declines in Latin America and worldwide (*Rosenblum et al., 2013*). This does not necessarily mean that only this lineage is present at these sites, as coexistence of Bd-GPL and other *Bd* strains has been shown at multiple sites using molecular data (*Byrne et al., 2019*). Genotyping of larger sample of Bd-positive amphibians from the region would be needed to understand whether Bd-GPL is the only strain present, and should be undertaken.

The failure of the *Bd* genotyping approach for our historical *Bd*-positive samples means that we cannot state with certainty that these older samples were infected with Bd-GPL. One of our current positive samples was assigned to Bd-GPL, and no other lineage of *Bd* has been reported from Mexico to date (*O'Hanlon et al., 2018*; *Byrne et al., 2019*). It is thus logical to consider that the lineage of *Bd* that we detected in 1969 may also have been Bd-GPL, but genotyping of additional contemporary *Bd* samples could reveal the presence of additional lineages, complicating this interpretation. Genotyping of our historic samples may have failed due to DNA damage from formalin fixation, possibly combined with a low amount of starting material as a result of low infection intensity. If specimens were fixed in formalin for a relatively long period of time (days) or stored in formalin prior to transfer to ethanol, this might decrease the probability of successfully amplifying fungal DNA from these specimens. Unfortunately, information on formalin fixation time is not available for the museum samples we attempted to genotype. The genotyping protocol has worked successfully with other museum samples (*Byrne et al., 2019*) but may need to be adapted to better amplify *Bd* from historical formalin-preserved samples. Genotyping additional historical positive samples will be useful to determine if preservation time or other factors impact the efficacy of the method in museum samples.

The qPCR assay we used has been shown to be sensitive to detecting Bd in formalin-fixed specimens that are decades old (*Cheng et al., 2011*; *Adams, Pessier & Briggs, 2017*). DNA damage (either from formalin or due to long preservation time) or low infection intensity could result in false negatives using this method, however, particularly in the oldest specimens. Because of this, we cannot exclude the possibility of false negative detections prior to the first *Bd*-positive specimen in 1969. Validation of historical *Bd*-positive and *Bd*-negative samples from qPCR through histological examination would strengthen the case for arrival of *Bd* at the end of the 1960s. It is also possible that some *Bd*-positive results from our qPCR assay resulted from cross contamination between specimens kept in the same jar in museums. The fact that most of the historical specimens in our sample tested negative for *Bd*, despite many being stored in jars with *Bd*-positive specimens, argues against this being a major problem, but it cannot be discounted with the sampling methodology we used.

The results of our field surveys across the range of *C. magnipes* complement *Bd* detection data in revealing apparent population extirpation at historical sites and low abundance in extant populations. We visited some sites only once during our survey and, given the low numbers of salamanders seen in extant populations and the fact that we did not always detect *C. magnipes* where it remains present, it is possible that salamanders are still present at other historical sites but went undetected. Prior to this study, however,

we (and others) have made multiple visits to Cueva de La Iglesia and Cueva de Potrerillos (the type locality of the species) without detecting *C. magnipes*. Very few individuals of *C. magnipes* were found in the two caves we surveyed with extant populations over the course of multiple visits. This stands in marked contrast to the dozens of individuals collected at historic sites before declines (Table 2). The low number of salamanders detected at these two sites could either be the result of ongoing small population size following declines due to epizootic *Bd* infection (or other causes) or a reflection that they naturally occur at low abundance at these sites. We never found more than five individuals on repeated visits to both sites from 2010 to the present, making it unlikely that the low number of individuals detected is a sampling artifact. The lack of precise estimates of *C. magnipes* abundance at historical sites (because not all individuals seen were necessarily collected) makes a direct comparison with previous abundance of the species impossible, but it is notable that neither of the contemporary population seems to reach the abundance reflected in museum samples from historic sites.

We note that almost all known localities of *C. magnipes* are along roads or near towns, where detection by local people or scientists is most likely; the karstic geology of the region means that there are likely hundreds or thousands of other small caves within more remote portions of the species' range that have not yet been searched. Furthermore, many parts of the caves we did explore cannot be accessed easily (at least not without professional equipment) and cave systems could be connected by subterranean passages, providing more habitat than is currently known. In most cases where we did find *C. magnipes*, they were relatively close to the mouths of caves, but we cannot discount the possibility that we were not able to detect salamanders in inaccessible or deeper portions of caves.

Taken together, these results suggest that the invasion of *Bd* may have played a role in the decline and apparent disappearance of *C. magnipes* from most known sites. Our data show that *Bd* was most likely absent from these populations prior to 1969. After that, it likely arrived several years before observed declines in salamander abundance, similar to declines in other Neotropical salamanders that were putatively caused by *Bd* (*Cheng et al., 2011*). Indeed, the lag time between first *Bd* detection and population declines and increases in prevalence of *Bd* seen in our results parallels that described in *Cheng et al. (2011)* from other sites in Mexico, although direct evidence linking *Bd* and declines at those sites is lacking. The fact that extant populations have low abundance of *C. magnipes* and are positive for *Bd* raises the possibility that infection at this site may have moved to an enzootic phase where *Bd* coexists with its host without causing mass mortality (*Briggs, Knapp & Vredenburg, 2010*).

While we document the arrival of *Bd* in populations of *C. magnipes* shortly before declines in host abundance, we lack data on the susceptibility of *C. magnipes* to *Bd* infection. Because *C. magnipes* is Critically Endangered and protected by Mexican law (Norma Mexicana 059-SEMARNAT-2010), it is not possible to collect individuals to conduct susceptibility trials for *Bd*. If *C. magnipes* is indeed highly susceptible to *Bd* infection, rapid elimination of infected individuals from the population could explain both the relatively low prevalence of *Bd* and the small number of individuals observed.

We cannot discount this possibility but we did not note any apparent signs of infection (loose skin, lethargy) when sampling individuals that later tested positive for *Bd*. Lacking information on susceptibility means that a direct link between *Bd* infection and salamander mortality cannot be established. Nevertheless, our data fit into a wider context of *Bd* invasion followed by collapse of host species and suggest that Bd-GPL may be implicated in the decline of *C. magnipes*.

While some of the caves surveyed had been damaged or were highly disturbed, most still appeared to contain appropriate conditions for salamanders, including wet surfaces, cracks, and invertebrate food sources. At other sites, however, habitat degradation may have played a role in observed declines. The type locality of *C. magnipes* at Potrerillos has little remaining vegetation around the cave mouth, likely affecting its microclimatic conditions to some extent. If *Bd* was responsible for declines of *C. magnipes*, it remains possible that declines in habitat quality played a role as well (separate from or synergistically with *Bd*). It is also possible that over-collecting at two sites (Potrerillos and Cueva de la Iglesia, Ahuacatlán), contributed to the decline of the species at these two sites. Large series of specimens were taken from these two sites between 1964 and 1975, possibly reducing abundance below the point where reproduction was sufficient to offset loss of individuals to museum collections. *C. magnipes* might be especially susceptible to over-collection, given that it occurs in spatially concentrated cave habitats. Over-collecting could not, however, account for the fact that the two extant sites surveyed (both recently discovered localities) have low abundance of individuals, despite never having had large series of specimens collected.

Our results add to a growing understanding of the widespread effects of the arrival of *Bd* in Mesoamerica by extending the detection of *Bd* back to 1969 in Mexico and to species that occur in specialized habitats such as caves. In Mexico, the previous earliest known *Bd* positive specimen came from Cerro Chicahuaxtla, Veracruz, approximately 345 km from our positive sample from 1969 (*Cheng et al., 2011*). *Chiropterotriton chico* collected in 1974 from El Chico National Park in Hidalgo, 130 km south of our 1969 *Bd*-positive sample, represents the nearest early *Bd* detection to the range of *C. magnipes*. The detection of *Bd* in *C. magnipes* to the north of previous historical positive samples is consistent with the hypothesis that Bd-GPL invaded from the north to the southeast through naïve amphibian populations (*Lips et al., 2008*; *Cheng et al., 2011*) in Mesoamerica. However, recent detection of *Bd*-positive amphibians from 1964 in Costa Rica (*De León et al., 2019*), as well as phylogenetic evidence for the presence of multiple lineages of *Bd* in other regions of Latin America (*Byrne et al., 2019*), suggest that *Bd* likely has a more complex history in the Americas than a simple north-to-south invasion. Successful genotyping of historical *Bd* positives from Mexico would clarify whether they resulted from a wave-like invasion of Bd-GPL or reflect the presence of a different lineage with a longer history in the region. Inherent properties of declining populations (low abundance during and after declines, inability to or restrictions on sampling the few remaining individuals) mean that the role of *Bd* in past declines of *C. magnipes* may never be definitively shown, but our results point towards a role for *Bd* in these declines.

The current conservation status of *C. magnipes* is challenging, given that nearly all known populations appear to be extirpated and both known extant populations are infected with *Bd*. The fact that individuals in both populations appear healthy and have maintained a relatively constant (if low) abundance since their discovery suggests that they are not in a state of continuing decline, but it is also possible that they could be negatively affected by *Bd*. In order to truly test this idea, dynamic data showing the host/pathogen dynamics through time are needed (*Briggs, Knapp & Vredenburg, 2010*; *Vredenburg et al., 2010*). If these populations are currently in an enzootic state of *Bd* infection, monitoring may be the best available option. Recent efforts to extirpate *Bd* in wild populations of amphibians via fungicide treatment (*Bosch et al., 2015*) were not successful in the long term. We suggest that monitoring both pathogen and host in this population of salamanders be implemented immediately to document changes in salamander abundance and infection intensity, which can be indicative of host survival (*Vredenburg et al., 2010*; *Lannoo et al., 2011*). Clearly, the cave habitats must be protected from excessive human visitation and other human impacts.

## CONCLUSIONS

Our results show that *Bd* may have contributed to the decline of one of the most enigmatic and unusual amphibians in the Neotropics. *Bd* appears to have arrived in 1969 after being previously absent, and its arrival occurred shortly before declines in abundance. Our genotyping results show that Bd-GPL, the lineage responsible for most amphibian declines, is currently present in populations of *C. magnipes*, but the failure to genotype historical samples means that we cannot definitively say that Bd-GPL was present in these populations in past decades. The fact that two known populations persist despite being infected with *Bd*, albeit at low abundance, suggests that populations may be able to coexist with the pathogen (*Knapp et al., 2016*) allowing them to persist into the future. At present, monitoring both host and pathogen may be the best available conservation option for this species.

## ACKNOWLEDGEMENTS

We thank Greg Schneider and Dan Rabosky (UMMZ), Toby Hibbits (TCWC), David Wake and Carol Spencer (MVZ), James Hanken (MCZ), Alan Resetar (FMNH), and Chris Austin (LSU) for access to museum collections and help with swab collection. Hassan Sulaeman provided critical help in the lab at SFSU, and Alexander Catenazzi kindly ran the qPCR assay for Bsal. Emanuel Martínez-Ugalde, Jorge Alejandro López Torres, María Guadalupe Segovia Ramírez, and Louis Paul Decena Segarra provided help in the field. Chris Grünwald provided the location of the La Trinidad population of *C. magnipes*.

### Funding

Funding was provided by CONACyT Problemas Nacionales Grant (Rovito, Grant #2015-721), Langebio-Cinvestav (Rovito), the National Science Foundation (Vredenburg,

Belmont Forum project NSF 163394) as part of the People, Pollution, and Pathogens Project (P3 project) through the call "Mountains as Sentinels of Change" by the Belmont-Forum (ANR-15-MASC-0001-P3, DFG-SCHM 3059/6-1, NERC-1633948 and NSFC-41661144004), and by the National Science Foundation (Rosenblum DEB-1457694). The funders had no role in study design, data collection and analysis, decision to publish, or preparation of the manuscript.

## Grant Disclosures

The following grant information was disclosed by the authors:
CONACyT Problemas Nacionales: #2015-721.
National Science Foundation: NSF 163394.
Belmont-Forum: ANR-15-MASC-0001-P3, DFG-SCHM 3059/6-1, NERC-1633948, and NSFC-41661144004.
National Science Foundation: DEB-1457694.

## Competing Interests

The authors declare that they have no competing interests.

## Author Contributions

- Mizraim Olivares-Miranda conceived and designed the experiments, performed the experiments, analyzed the data, prepared figures and/or tables, authored or reviewed drafts of the paper, and approved the final draft.
- Vance T. Vredenburg conceived and designed the experiments, authored or reviewed drafts of the paper, and approved the final draft.
- Julio C. García-Sánchez conceived and designed the experiments, performed the experiments, analyzed the data, prepared figures and/or tables, authored or reviewed drafts of the paper, and approved the final draft.
- Allison Q. Byrne conceived and designed the experiments, performed the experiments, analyzed the data, prepared figures and/or tables, authored or reviewed drafts of the paper, and approved the final draft.
- Erica B. Rosenblum conceived and designed the experiments, authored or reviewed drafts of the paper, and approved the final draft.
- Sean M. Rovito conceived and designed the experiments, performed the experiments, analyzed the data, prepared figures and/or tables, authored or reviewed drafts of the paper, and approved the final draft.

## Animal Ethics

The following information was supplied relating to ethical approvals (i.e., approving body and any reference numbers):

Centro de Investigación y de Estudios Avanzados del Instituto Politécnico Nacional provided approval for this research (SIACUAL permit 0291-19).

## Field Study Permissions

The following information was supplied relating to field study approvals (i.e., approving body and any reference numbers):

Field experiments were approved by the Secretaria del Medio Ambiente y Recursos Naturales, Mexico (SGPA/DGVS/002764/18).

## DNA Deposition

The following information was supplied regarding the deposition of DNA sequences:

The multilocus sequence data used for Bd genotyping are available at the NCBI Short Read Archive BioProject: PRJNA608001.

## Data Availability

Data for chytridiomycosis qPCR assays (specimen data and assay results) are available as a Supplemental File and deposited in AmphibiaWeb's Amphibian Disease Portal under the project name "Fungal infection, decline, and persistence in the only obligate troglodytic Neotropical salamander" (https://amphibiandisease.org/projects/?id=290).

## Supplemental Information

Supplemental information for this article can be found online at http://dx.doi.org/10.7717/peerj.9763#supplemental-information.

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
