# Peer review of "Fungal infection, decline and persistence in the only obligate troglodytic Neotropical salamander"

_PeerJ, doi:10.7717/peerj.9763_

## Round 0.1 · original submission · Minor Revisions

Thank you for submitting your manuscript to PeerJ. I have sent your paper to expert referees for their consideration. I have now received their comments back and have read through your paper carefully myself. Enclosed please find the reviews of your manuscript.
The reviews are in general favourable and suggest that, subject to minor revisions, your paper could be suitable for publication. Please carefully consider these suggestions and address all the questions raised on point by point basis, and I look forward to receiving your revision.

·

Basic reporting

Olivares-Miranda et al. explored whether the chytrid fungus Batrachochytrium dendrobatidis (Bd) could be implicated in declines of Chiropterotriton magnipes, the only obligate troglodytic Neotropical salamander. To this aim, the authors used molecular techniques to detect Bd in both extant and historical (preserved) specimens, conducted field surveys, and attempted to identify the present Bd lineage(s) through genotyping. The authors use clear and unambiguous, professional English throughout. The introduction does a great job in summarizing knowledge on emergence and spread of Bd in Latin America, clearly identifies knowledge gaps, and introduces the case study in a way that emphasises its relevance within the context of amphibian disease decline research. Overall, the manuscript is well-structured, self-contained, and includes necessary info on which ethical body approved the research, and the institute that issued field permits.

I suggest that the authors move the sentence at lines 69-72 ('Declines occurred - Vredenburg et al 2019') upwards, to line 64 directly after 'Catenazzi et al. 2011).'. This way, it is immediately clear why there are Bd records in Costa Rica that predate Bd epizootics, and why Bd was only described in 1999 (lines 64-69). I then suggest to to remove lines 72-79, as the information in these lines is largely repeated in paragraph 80-95. Part of the information in lines 72-79 (for instance, the interesting record from Bolivia) can be integrated into the next paragraph (e.g. around lines 88-93).

Lines 263-268: This section lacks necessary information. If I understand well from Table 3, four magnipes (not two) from two populations tested positive for Bd, is this correct? Also, from which population was Bd genotyped? I was unable to find this information in the text or tables. The sentence ‘to another sample collected from Chiapas’ (line 265) implies that the genotyped sample originated from an Ixalotriton or Nyctanolis population (although I do understand from Table 3 that this isn’t the case). Please rewrite this paragraph, and move sentence 266-267 to the discussion.

Table 1: The use of ‘sampling size’ in the caption title, ‘sampling results’ in the caption, and ‘Number of C. magnipes detected’ in the table itself is confusing. I assume that the latter is correct; i.e., the number displayed in the last column is the amount of magnipes that you observed, but not all could be captured for sampling, right? Please clarify and change the caption.

Table 3: Please add more descriptive information in the caption; for instance, do I understand well that the first ‘N’ column refers to the number of swabs tested for Bd, while the second refers the number of swabs tested for Bsal? Could you explain why these numbers occasionally differ?

Figure 1 is relevant and well-described, although seems to be of low resolution – but this might be a result of automatic downsizing while setting up the review pdf. If this is not the case, please provide a higher resolution version.

The data deposition statement contains information and external links to the multilocus sequence data used for Bd genotyping and data for chytridiomycosis qPCR assays (which are working, all data is available). The deposition information however seems to be missing from the manuscript – please add.
Please note that the data for chytridiomycosis qPCR assays still contains high-resolution locality data for all sampled populations.

Experimental design

In their study, the authors attempt to answer if (arrival of) Bd was implicated in declines of C. magnipes. This is both an important and timely question, especially in the light of recent discussions on the strength and availability of (retrospective) evidence linking chytridiomycosis to amphibian declines, in which the authors partook. The current study adds much-needed information on local presence and estimated prevalence of Bd across time, and is therefore an important contribution to the field.

The employed methods are state-of-the-art, and either follow recently published protocols or are sufficiently described to allow replication. I particularly liked to see that all qPCR analyses were conducted in duplicate or triplicate, and included both positive and negative controls.

Validity of the findings

As mentioned by the authors, the information presented in the manuscript points to a role of Bd in driving salamander declines, but does not provide unambiguous evidence, in part due to low Bd prevalence during declines, limited sample sizes in some time periods, and failure to genotype Bd from historical material. The authors weigh their findings accordingly in the discussion, and do not overstate their results. Larger samples sizes, especially from contemporary times, would have allowed more accurate estimates of infection intensity and prevalence – but I’m aware that these are difficult to obtain due to low abundance of many salamanders in Mesoamerica nowadays. While taking these unavoidable limitations into account, the authors draw sound conclusions, which (as I mentioned before) provide valuable information on local Bd presence and estimated prevalence across time.

Additional comments

Line 19 Batrachochytium > Batrachochytrium
Line 80: It would be good to specify that this concerns Bd-GPL.
Line 101: remove 'most of which belong the genus Chiropterotriton'
Line 148 Bd not in italics
Line 174 Bd not in italics
Lines 175-176: Add a few words to make clear that Ixalotriton and Nyctanolis do not only live in cave environments – it is currently confusing that these are introduced as cave salamanders, while the introduction states that C. magnipes is the only Mexican cave salamander.
Line 187 Bsal not in italics
Lines 189-190: Please add the number of positive and negative controls
Line 205 Bd not in italics
Line 207 (Byrne et al. 2017) > Byrne et al. (2017).
Line 230 Bd and Bsal not in italics
Line 231 calculate > calculated
Line 233 analysis > analyses
Line 243 > remove ‘in’
Line 244 Bd not in italics
Line 245 Bd not in italics
Line 253-255 replace with ‘We detected Bd on other amphibians at three of the six sites where these were found (Table 3), of with 25% (8/32) tested positive’.
Line 256 El Coni or Durango (latter was used earlier in the text)?
Line 259 Bd not in italics
Line 260 Table 1 > Table 3; Table 1 does not contain Bd presence of prevalence data
Line 277 Bd not in italics
Line 281 Bd not in italics
Line 290 Bd not in italics
Line 295 Table 1 > Table 3; Table 1 does not contain Bd presence of prevalence data
Line 296-298 El Coni or Durango? Please check throughout the manuscript.
Line 300 Craugastor decorates > decoratus
Line 307, 308 Bd not in italics

Reviewer 2 ·

Basic reporting

no comment

Experimental design

no comment, please see general comments to authors

Validity of the findings

For specific comments, please see general comments to authors.

Additional comments

The authors present results of a study of a rare Mexican cave salamander, focusing on the potential involvement of chytrid fungus (further: Bd) in the species’ decline. Overall, I enjoyed reading this manuscript, which presents novel and relevant (be it mainly for a single species) information. The relative paucity of information presented is due to the rarity of the species; the entire group of neotropical plethodontids has been largely neglected by science (apart from some very active scientists including David Wake and some of the authors eg Sean Rovito) and every bit of new information is valuable from a conservation perspective. I do have some concerns, which I feel should be fairly easy to address and, although these are relatively minor, I would encourage the authors to include these in their revised version.
1) line 76: this is one of my main remarks throughout the manuscript: a major limitation of most studies that include museum specimen of amphibians to be examined for the presence of Bd, is that they are based on swab-sampling specimens kept in jars, which often contain multiple specimen (sometimes multiple species). Although rinsing animals before sampling will remove some contamination, cross-contamination from other specimen in the same jar (and even at the time of collection: between jars) should never be excluded. This would require additional techniques such as histology or (preferably) immunohistochemistry. The claim of finding a Bd positive specimen in 1863 should be considered in the light of potential contamination. I would like to see this nuance added in the introduction. For the results of this paper: it would be appropriate to taken this into account as well. This will not have any major impact on the relevance of the paper, or its conclusions. I recommend adding information on number of specimen per jar examined and the species present, and a brief comment in the discussion with regard to estimating prevalence based on mere swab sampling museum specimen. I agree with the authors that the sampling method renders estimating infection loads rather unreliable but I would still recommend providing GE counts for the positive samples: it is important to mention this ao to estimate the probability of contamination.
2) My second major remark is that it is unfortunate that sampling of the museum animals was restricted to collecting swabs. If possible, efforts should be made to try to perform histopathology on the animals that tested positive. This would provide very valuable information with regard to 1 pathological effects on the skin (eg any signs of pathology such as hyperplasia and/or hyperkeratosis) and 2 independent confirmation of actual infection (not contamination of the animals).
3) A third somewhat major issue is the possibility of repeated samplings of the same animals in the two caves: were these all different animals and, if so, how was this confirmed? Given the low density, possibly (?) small home range of the animals, this may be an issue and would affect any estimate of eg prevalence.
4) I fully understand few data are available to be more objective regarding the species’ decline. This uncertainty, however, should be reflected in statements throughout the text (eg Line 108: “greatly reduced abundance”, this implies data to be present).
5) The authors should comment on Bd detectability in formalin fixed specimen. Certainly low infection loads in formalin fixed specimen may be prone to result in false negatives. I agree with the authors that recent Bd incursion (late 60’s) is the most plausible explanation, but their data (small sample size, long term formalin fixed animals) do not allow robust conclusions with regard to excluding a different scenario: long term endemism at low prevalence and low infection loads before 69, with subsequent increase (endemic hypothesis) cannot be excluded based on their data. This may seem rather unlikely but at least needs mentioning in the discussion.
6) The authors should comment on detectability of Bd in relation to the time the sample has been stored in formalin. If no information is available, this should be mentioned and discussed (mentioning implications for conclusions).
7) I would strongly advise against including the other (non syntopic) Mexican cave salamanders (Ixalotriton and the stunning Nyctanolis): sample size is too small to allow any sound conclusions and it does not really add anything significant to the manuscript.
8) line 180: what is meant by: “all samples were processed immediately before release”?
9) In the results section, please mention the number of samples obtained (not only for the C. magnipes).
10) The authors demonstrate Bd presence on living animals, at a fair prevalence of 29% and infection loads that are low to relatively high. I have several suggestions here:
a. please provide information on life stage / sex of the salamanders sampled. Life stage dependent infection and disease dynamics of Bd have been demonstrated (eg it may be possible that Bd exerts negative effects mainly on juveniles, which would impair recruitment)
b. Stating that infection loads are low should be revised. At least one sample showed a GE load of 2618. While this is indeed not a load one would associate with lethal infections, I would recommend some caution here: the authors have no isolate and the number of GE is a calculation. The actual ITS copy number is important to make any statement with regard to absolute number on the swab.
c. please discuss the possibility that infected animals are quickly eliminated from the population (as an explanation for the relatively low prevalence).
11) Table 2: please add locality information here.
12) Please refrain from any conclusion based on very small sample sizes (eg line 287-289, line 298-299)
13) Please be careful with any conclusions regarding the species absence in the caves examined. The information provided in Table 1 shows that the majority of caves was visited only once (locality 6 is mentioned twice). Looking at both localities were animals were found (7, 13), numbers are low and even 0 on two occasions.
14) A bit in line with the previous remarks: please comment on potential connectivity between the different cave systems and discuss in light of Bd epidemiology. I have no personal experience with these species, but I do have with many palearctic species with a (largely) subterraneous ecology. Caves do provide access to for example carstic limestone underground, but the vast majority of these systems is inaccessible for humans (except for the larger cave systems). Would it be correct to state that the species may have a vastly more broad distribution range than only those few caves? Please briefly discuss.
15) Table 1: please add information per cave: terra typica, availability of historical data (C magnipes presence), newly searched caves.
16) Table 3: I would recommend to limit to those localities, relevant for C magnipes.

---

## Round 0.2 · accepted · Accept

Thank you for taking the time to revise and resubmit your manuscript. I have now read through your paper as well as your letter in response to the reviews. I think that you have successfully addressed all of the concerns raised very well, and would like to accept your manuscript for publication in PeerJ.

Congratulations!

Thank you for all the hard work you have put into this. Your paper makes a strong contribution to the literature and I look forward to seeing it published. All the best and stay safe.